# The association between adolescents' independent food purchasing and dietary quality differs by socioeconomic status: Findings from a pilot study

Sarah Shaw[1,2¤]*, Sarah Crozier[1,3], Cyrus Cooper[1,2], Dianna Smith[3,4], Mary Barker[1,2,5], Christina Vogel[1,2,3,6]

**1** MRC Lifecourse Epidemiology Centre, University of Southampton, Southampton, United Kingdom, **2** NIHR Southampton Biomedical Research Centre, University of Southampton and University Hospital Southampton NHS Foundation Trust, Southampton, United Kingdom, **3** NIHR Applied Research Collaboration Wessex, Southampton Science Park, Innovation Centre, Southampton, United Kingdom, **4** Geography and Environmental Science, University of Southampton, Southampton, United Kingdom, **5** School of Health Sciences, Faculty of Environmental and Life Sciences, University of Southampton, Southampton, United Kingdom, **6** Centre for Food Policy, City St Georges, University of London, London, United Kingdom

¤ Current address: MRC Epidemiology Unit, University of Cambridge, Institute of Medical Science, Addenbrooke's Hospital, Cambridge, United Kingdom
* ss@mrc.soton.ac.uk, sarah.shaw@mrc-epid.cam.ac.uk

## Abstract

During adolescence, many young people start to make more independent food purchases. Subsequently, these independent food choices will increasingly contribute to their overall diet quality; little is known, however, about this relationship. This pilot study aimed to examine the role adolescents' independent food purchases play in their diet quality and assess if these relationships vary according to socio-economic status. A convenience sample of adolescents aged 11–18 years and attending secondary school or college in Hampshire, England, were recruited to participate in a one-week cross-sectional observational study. A validated 20-item Food Frequency Questionnaire assessed diet quality. Participants used an ecological momentary assessment mobile phone app to record food purchases. Over seven days, 552 food/drink items were purchased on 253 food purchasing occasions by 80 participants. The majority of purchases (n = 329, 59%) were coded as 'not adhering' to the UK Eatwell Guide, 32% were coded as 'adhering' and 9% fell between these categories being coded as 'combination'. The healthfulness of food purchases did not differ between adolescents from low- and high-SES households. Across all adolescents, 39% reported that their food and drink purchases were snacks and these were less healthful than purchases made for main meals. Fully adjusted regression models showed that adolescents who made less healthy food purchases tended to have poorer quality diets (β 0.36, (95%CI −0.15, 0.87) p = 0.16). Interaction models showed that less healthy purchasing was more strongly associated with poorer diet

**Data availability statement:** Data cannot be shared publicly because of ethical restrictions imposed in the interest of protecting the anonymity of the study participants who are aged 18 years and under. Data are available from the University of Southampton data repository (contact via researchdata@soton.ac.uk) for researchers who meet the criteria for access to confidential data.

**Funding:** This research and the authors of this paper are supported by the following funding sources: National Institute for Health Research Southampton Biomedical Research Centre; UK National Institute for Health Research Programme Grants for Applied Research (RP-PG-0216-20004); and UK Medical Research Council (MC_UU_12011/4). The views expressed in this publication are those of the author(s) and not necessarily those of the National Health Service, the National Institute for Health Research, and the UK Department of Health and Social Care. The funders had no role in study design, data collection and analysis, decision to publish, or preparation of the manuscript.

**Competing interests:** SS, SC, CC, DS, MB have no conflicts of interests to declare. CV has a non-financial research collaboration with a UK supermarket chain. The study described in this manuscript is not related to this relationship. This does not alter our adherence to PLOS ONE policies on sharing data and materials.

quality among young people of lower SES than those of higher SES (p = 0.01). Future research should focus on identifying ways to support more healthful independent food choices by adolescents to reduce dietary inequalities and improve health and well-being among the next generation of adults.

## Introduction

Poor dietary behaviours are a major contributor to the global burden presented by non-communicable diseases (NCDs) [1]. Evidence from the UK's National Diet and Nutrition Survey suggests that adolescents have poorer dietary behaviours when compared to other age groups [2,3]. The dietary habits of this age group are an important public health issue as behaviours established during this period frequently track into adulthood and play a role in future health trajectories [4]. Identifying ways to support healthy dietary behaviours among adolescents is therefore an important strategy to improve health and reduce the prevalence of NCDs; intervening in adolescence has potential to improve the immediate and future health of the individual as well as the health of any future offspring [5].

Adolescence is a time when individuals gain autonomy and independence. During this stage of the lifecourse, many young people start making more of their own food choices without supervision from parents or other care givers. While it has been shown that adolescents source the majority of their food from home and school [6–8], many adolescent autonomous food decisions are likely to be made outside of these home and school environments. Few studies have provided detailed information about the types of foods adolescents purchasing for themselves. A 2015 report from Food Standards Scotland showed that the majority of independent food choices made by adolescents outside of the home and school at lunchtimes are high in fat, sugar and salt; chips (fries), sandwiches, sugary soft drinks and energy drinks were some of the most frequently purchased items [9]. Independent food choices made by adolescents are likely to represent an increasing proportion of their overall food intake and, therefore, their overall diet quality. There is limited evidence, however, about how the foods adolescents purchase for themselves affect their overall dietary quality. Findings from a study in Kentucky, USA, showed that adolescents with unhealthy shopping habits (shopping three or more times per week at petrol stations, convenience stores and fast-food restaurants) ate fewer fruits and vegetables than those with healthier shopping habits [10]. Understanding the contribution that independent food purchases of adolescents make to their overall dietary intake is important to design interventions and food policies that are effective at improving the diet and health of this important age group.

Socio-economic status (SES) has been well documented as a critical determinant of dietary quality. In the UK, adolescents living in more deprived areas and lower income households have been shown to have poorer quality diets [11–13], in particular consuming lower levels of vegetables, fibre and oily fish and more energy drinks

[11–13]. However, the role that adolescents' independent food purchasing decisions play in their overall diet quality and the difference by SES remains unclear.

In order to address this knowledge gap, this study aimed to:

1) Examine food purchasing behaviours as predictors of diet quality;

2) Assess whether SES moderated the relationship between food purchasing behaviours and diet quality of adolescents.

## Materials and methods

### Study design

A one-week cross-sectional, observational pilot study was conducted to collect data on adolescents' independent food purchasing behaviours and diet quality as part of the Map My Food Study. This study was conducted according to the guidelines laid down in the Declaration of Helsinki and all procedures involving research participants were approved by the Faculty of Medicine Ethics Committee, University of Southampton (ERGO 57044).

### Participants and recruitment

A convenience sample of adolescent participants was recruited from schools, colleges and youth groups based in Hampshire in the south of England, UK. Adolescents were eligible to participate if they i) were aged 11–18 years, ii) attended school or college in England and iii) owned a GPS-enabled smart phone.

Details about the study, including participant information sheets, were circulated to potential participants by announcements and student emails at the participating organisations. A researcher followed up in-person, when possible, by attending the school, college or youth group to provide further details and answer questions from the young people. Information (both written and verbal) was provided about the purpose and methods of the study. Written informed consent was obtained for all participants using an electronic consent form. Participants over the age of 16 years were able to provide consent to participate. Those aged under 16 years provided assent and parental consent was also sought for their participation.

### Data collection

Data were collected between May 2021 and April 2022. Following the consent process, participants were asked to complete an online questionnaire containing demographic questions and a food frequency questionnaire (FFQ). Participants were also asked to download and use a mobile phone app for one week. The app was a modified version of the Tripzoom app which was created and run by a commercial company, Locatienet [14,15]. A member of the research team aimed to be present when the participants were completing the questionnaire and downloading the study app. On some occasions, restrictions on visitors entering schools and colleges during the COVID-19 pandemic prevented this happening. In these instances, participants were able to contact the research team via email if they experienced any difficulties. A researcher was present for 59% of app downloads.

The app used Ecological Momentary Assessment (EMA) methods to collect data about the food outlets adolescents used and their food purchases over the study week. EMA is an intensive longitudinal data collection technique aimed at reducing recall bias by collecting data from study participants when they are in their natural environment, and close in time to when the behaviour of interest takes place [16]. EMA has been used previously with adolescents and young adults in order to provide a better understanding of the external and internal cues influencing dietary consumption [17–20]. Adolescents were asked to use the app to record all their independent food and beverage purchases over the study week. Three time-based prompts were delivered through the app to remind participants to make a recording and to reduce incidences of missing data. To make a recording, adolescents could select a food or drink from a pre-defined list of food items. If an

item was not present on the list, details could be entered using a free text field. Participants could also take photos of the items or receipts. These photos were used to check the accuracy of the recorded food items. In total, photos were provided for 23% of entries. With each entry, participants were also asked to provide details about the purpose of the purchase (breakfast, lunch, dinner or snack). The app data collection methods were refined through feasibility testing (n = 14) and a pre-pilot study (n = 12) with young people to ensure the number of prompts as well as the wording and quantity of questions were acceptable [21]. Following the pre-pilot study, 83% of participants responded *neutral*, *agree*, or *strongly agree* to the statement, *"I liked using the app."* However, 42% reported that there were too many notifications, leading to a reduction in the number of prompts to three per day in the full study. At the end of the study week, participants who had completed the questionnaire and downloaded the app were given a £10 Amazon voucher as an appreciation for their participation in the study.

### Exposure measures

**Demographic data.** Using the questionnaire, participants provided their age, gender, ethnicity and home postcode. Household-level SES was assessed using questions from the Family Affluence Scale (FAS) [22] (Inchley et al. 2018). These questions have been used in the UK-based 'What about YOUth?' Survey [23] and the International Health Behaviour in School-aged Children study [22]. The questions have been validated and are shown to be easier to complete than questions about parental education and occupation [22]. A binary score was created to represent household SES: low household SES (FAS 1–7), high household SES (FAS 8–14). The Index of Multiple Deprivation (IMD) [24], the official measure of relative deprivation for small geographical areas in England, was calculated for each participant using their home postcode. A binary score was created to represent neighbourhood SES: low neighbourhood SES (IMD scores 1–5), high neighbourhood SES (IMD scores 6–10).

**Total number of food purchases.** Using the food purchasing data recorded in the app, the number of food/drink items purchased over the study week was tallied for each participant. First, the number of items purchased on each food purchasing occasion was tallied; for example, if a participant purchased a burger, chips, and a sugar sweetened beverage, three items would be recorded for that food purchasing occasion. These numbers were then tallied for all food purchasing occasions made by each participant to give the total number of food purchases made over the study week.

**Healthfulness of food purchases.** All food and drink items purchased during the Map My Food Study were classified according to how well they adhered to the UK's Eatwell Guide [25]. Food and drink items were classified as either 'adhering', 'not-adhering' or 'combination'. Food and beverage items which are included as part of the main food groups in the Eatwell Guide and those that individuals are encouraged to eat more were categorised as 'adhering' (e.g., fruit, non-sugar sweetened drinks, bread). Foods high in fat, sugar and salt that fall outside the main food groups in the Eatwell Guide and which individuals are encouraged to limit were categorised as 'not-adhering' (e.g., crisps, sugar-sweetened beverages, confectionary). Foods that could not be classified within either of these groups were classified as 'combination' foods (e.g., sandwich/wraps, hot drinks (not tea or coffee). Supporting Information Table 1 shows the food purchases recorded by participants and how they were classified in terms of healthfulness. Previous research has used similar coding approaches based on the Australian Guide to Healthy Eating [6,26].

To create Purchasing Healthfulness Scores, foods classified as 'adhering' were allocated the score of '+1', the 'combination' classification was allocated the rating of '0' and the 'non-adhering' foods were allocated the rating of '-1'. Food scores were subsequently tallied for each food purchasing occasion and then divided by the total number of food/drink items purchased as part of that food purchasing occasion. The same process was used to calculate a Weekly Purchasing Healthfulness Score for each participant; food scores were tallied for the entire week and then divided by the total number of food/drink items purchased over the week. Scores ranged between −1 and 1 and were used as continuous variables and presented as healthfulness units. Higher scores representing healthier independent purchasing patterns.

 

**Table 1. Characteristics of study participants.**

| Characteristics | Participants with purchasing and diet data (n = 80) | Participants without purchasing data (n = 28) | p-value |
|---|---|---|---|
| **Gender, n (%)** | | | 0.03 |
| Boy | 19 (24) | 14 (50) | |
| Girl | 55 (69) | 13 (46) | |
| Not provided | 6 (7) | 1 (3) | |
| **Age, median (IQR) (years)** | 17 (15,17) | 14 (13,17) | <0.01 |
| **Age, n (%) (years)** | | | <0.01 |
| 11-13 years | 8 (10) | 9 (32) | |
| 14-16 years | 25 (31) | 10 (36) | |
| 17-18 years | 47 (59) | 9 (32) | |
| **Ethnicity, n (%)** | | | 0.22 |
| White | 63 (79) | 25 (89) | |
| Other ethnicities* | 17 (21) | 3 (11) | |
| **Area-level SES (IMD), n (%)** | | | 0.92 |
| 1-5 (most deprived) | 30 (37) | 12 (43) | |
| 6-10 (least deprived) | 34 (43) | 13 (46) | |
| Missing | 16 (20) | 3 (11) | |
| **Household-level SES (FAS), n (%)** | | | 0.21 |
| Low (0–7) | 24 (30) | 5 (18) | |
| High (8–14) | 54 (68) | 23 (82) | |
| Missing | 2 (2) | 0 (0) | |

* Other ethnicities: Black African, Bangladeshi, Pakistani, Chinese, Afghan, British Indian, British Punjabi, Arabic, British Asian, Turkish, Mixed

## Outcome measures

**Diet quality.** Using a 20-item FFQ that was designed to assess diet quality among UK adolescents in population level studies [11], participants recorded how frequently they consumed each food group over the previous month. These dietary data were used to calculate a diet quality score for each participant following published methodology specifically developed to assess diet quality among UK adolescents [11]. This diet quality score has been validated against fourteen nutritional biomarkers, including serum folate, homocysteine, total carotenoids and vitamins B12, C and D [11]. Diet quality scores were used in analyses as a continuous variable with higher scores representing better quality diets, consistent with the UK dietary guidelines.

## Statistical analysis

The normally-distributed continuous variable (diet quality) was summarised using mean (SD). Non-normally distributed continuous variables (age, neighbourhood SES, household SES are summarised using median (IQR). Categorical variables are summarised using n (%). T-tests, Mann-Whitney rank sum tests and Chi-squared tests were used to compare differences in demographic characteristics between those with purchasing data and those without. Differences in diet quality scores and purchasing outcomes according to demographic characteristics were assessed using unpaired t-tests and Mann-Whitney rank-sum tests. In order to retain statistical power, a Spearman's correlation test was conducted for characteristics measured using an underlying continuous variable (age, IMD, and household SES).

To address aim 1, two separate linear regression models were fitted both using the Diet Quality Score as the outcome. The exposure for the first regression model was the variable describing the number of food purchases over the study week. The exposure for the second model was the Weekly Purchasing Healthfulness Score. Age, gender, ethnicity, and household SES were included as confounding variables in adjusted models, reflecting their observed influence in previous studies [11,27]. In recognition that scoring food purchases categorised as 'combination' as '0' may not be capturing the missing information associated with these foods, a sensitivity analysis was conducted with these purchases treated as missing. This sensitivity analyses showed no impact on the overall study findings. Full details of the sensitivity analysis can be found in the Supporting Information.

To address aim 2, an interaction term for Weekly Purchasing Healthfulness Score by household SES was added to the regression models to determine the effect modification of household SES in the relationship between food purchasing behaviours and diet quality.

Regression model results are presented with effect sizes and confidence intervals. Standard errors can be calculated using the equation $\beta + (1.96 \times SE)$. Due to the pilot nature of the study, the results were interpreted based on effect sizes rather than emphasising statistical significance; this stance is in line with current statistical thinking on reporting findings from quantitative studies in medicine [28]. Regression diagnostics were assessed for all models and found to be satisfactory. All data were analysed in Stata version 16 [29].

## Results

### Participant characteristics

A total of 108 participants completed the online questionnaire. Of these, 101 participants downloaded the study app and food purchasing data were available for 80 participants. Table 1 shows the demographic characteristics for participants with and without purchasing data. The majority of participants with both purchasing and dietary data were girls (69%), self-identified as being of white ethnicity (79%) and had a median age of 17 years (IQR 15, 17). Thirty participants (37%) were living in homes located in the most deprived half of neighbourhoods in the UK. Twenty-four participants (30%) had a household SES (Family Affluence Score) between 0–7, representing low household-level SES. Boys (p = 0.03) and younger adolescents (p < 0.01) were more likely to have no purchasing data.

### Food purchasing behaviours

In total, 552 food/drink items were purchased during 253 food purchasing occasions over the study week. The median number of purchases made by each participant over the 7-day period was 5 (IQR 3, 8; range 1–30). The most frequently purchased foods over the study period were chips (n = 47, 9% of all purchases), sandwiches (n = 40, 7% of all purchases) and vegetables (n = 30, 5% of all purchases) (Fig 1). The most frequently purchased drinks were non-sugar-sweetened carbonated drinks (n = 28, 5% of all purchases), sugar-sweetened carbonated drinks (n = 24, 4% of all purchases) and water (n = 19, 3% of all purchases). The majority of purchases (n = 329, 59%) were coded as 'not adhering' to the UK Eatwell Guide, 175 food items (32%) were coded as 'adhering' and 48 (9%) were coded as 'combination' foods.

Similar purchasing patterns were shown between adolescents from high and low SES households. For adolescents from high SES households, 214 food items (60%) were coded as 'not adhering', 112 (31%) were coded as 'adhering' and 32 (9%) were coded as intermediate foods. For adolescents from low SES households, 115 food items (59%) were coded as 'not adhering', 63 (33%) were coded as 'adhering' and 16 (8%) were coded as 'combination' foods.

Of the 253 food purchasing occasions, participants reported that the food and drinks purchased on 39% of these occasions were for snacks, 26% were for lunch, 26% for dinner and only 7% were for breakfast (Table 2). Purchases bought for snacks were less healthy than those made for breakfast, lunch or dinner (p < 0.001).

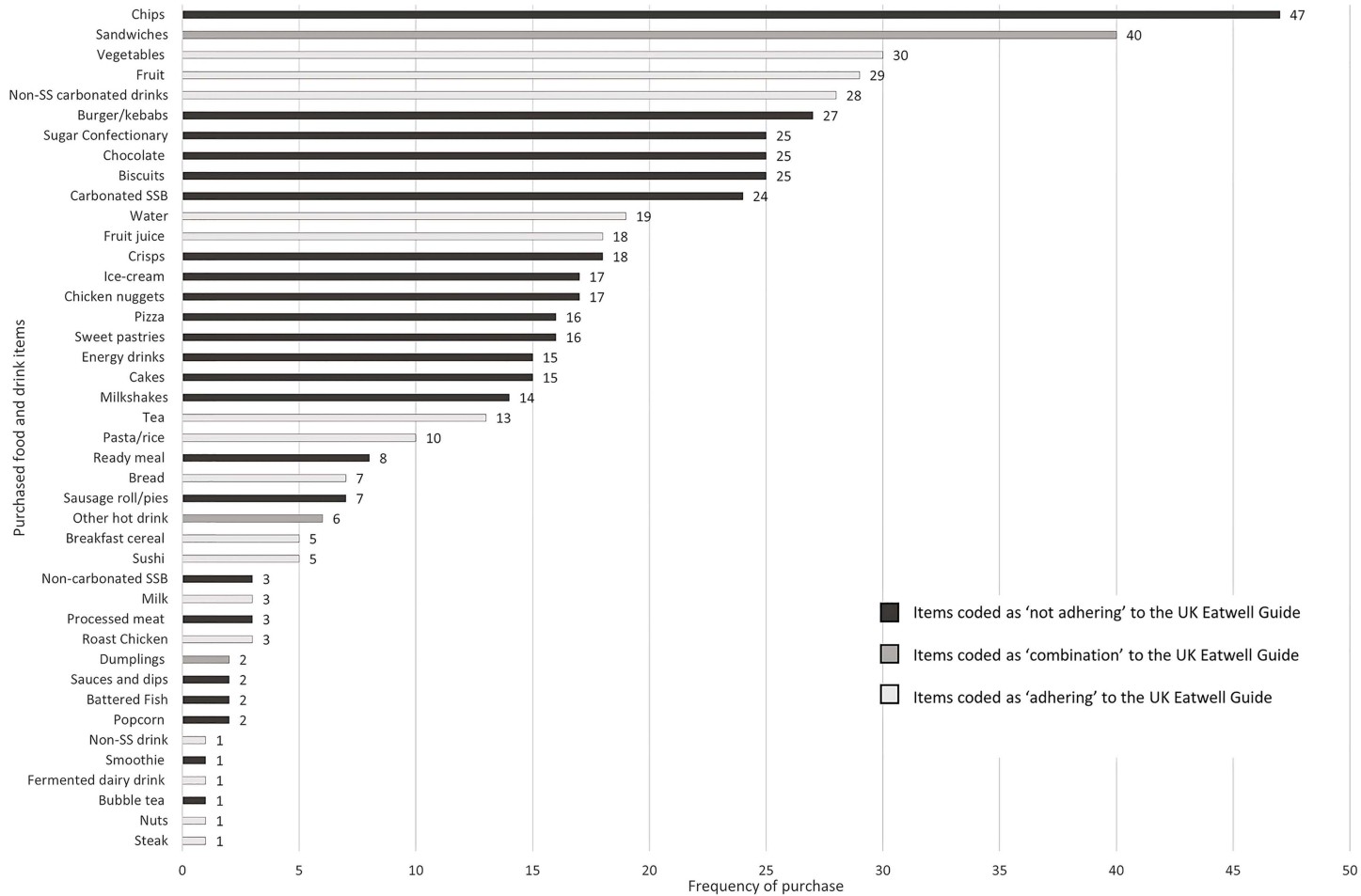

**Fig 1. Graph illustrating the types and frequency of foods/drinks purchased by adolescents during the study.**

The median Weekly Purchasing Healthfulness Score, reflecting the overall healthfulness of all purchases made over the study week, was −0.33 healthfulness units (IQR −0.65, 0.00) indicating that for every healthy purchase made, on average two unhealthy purchases were made. Table 3 presents the median number of weekly purchases and median Weekly Purchasing Healthfulness Score according to participants' demographic characteristics. On average, older adolescents and girls purchased more food and drink items over the study week compared to younger adolescents (p = 0.07) and boys, respectively (p = 0.003). Adolescents of white ethnicity tended to purchase more items than those of other ethnicities (p = 0.22). Participants from more deprived backgrounds purchased more food and drink items than those from more affluent backgrounds (IMD p = 0.20, FAS p = 0.71). No notable differences were observed in Weekly Purchasing Healthfulness Score according to age, gender, ethnicity, and SES measures.

### Diet quality

Table 3 shows mean Diet Quality Scores according to demographic characteristics. Diet Quality Scores were similar across the three age groups but tended to be higher among girls compared to boys (p = 0.29) and higher among adolescents of white ethnicity compared to other ethnicities (p = 0.09). Diet Quality Scores were also higher among adolescents

**Table 2. Purchasing Healthfulness Scores on food purchasing occasions according to purpose of purchase.**

| Purpose of purchase | n | (%)* | Purchasing Healthfulness Score[a] (Median (IQR)) | p-value |
|---|---|---|---|---|
| Breakfast | 18 | (7) | 0.00 (−0.33, 0.50) | <0.001[b] |
| Lunch | 66 | (26) | 0.00 (−0.67, 0.33) | |
| Dinner | 66 | (26) | −0.25 (−1.0, 0.00) | |
| Snack | 98 | (39) | −1.00, (−1.0, 0.00) | |
| Missing | 5 | (2) | – | |

*% of food purchasing occasions (n total = 253)

[a]Purchasing healthfulness score on food purchasing occasions

[b]Kruskal-Wallis test with adjustment for ties

Purchasing Healthfulness Scores closer to 1 indicate healthier purchasing

**Table 3. Participant purchases and diet quality scores according to characteristics.**

| Participant Characteristics | Number of weekly purchases (Median (IQR)) | n (n=80) | p-value | Weekly Purchasing Healthfulness Score (Median (IQR)) | n (n=80) | p-value | Diet Quality Score (Mean (SD)) | n (n=108) | p-value |
|---|---|---|---|---|---|---|---|---|---|
| **Age** | | | 0.07[a] | | | 0.72[a] | | | 0.77[a] |
| 11-13 years | 4.50 (2.50, 5.50) | 8 | | −0.17 (0.63, 0.00) | 8 | | −0.05 (0.75) | 17 | |
| 14-16 years | 5.00 (3.00, 9.00) | 25 | | −0.33 (−0.88, 0.00) | 25 | | −0.07 (1.18) | 35 | |
| 17-18 years | 6.00 (4.00, 9.00) | 47 | | −0.33 (−0.60, 0.00) | 47 | | 0.06 (0.96) | 56 | |
| **Gender** | | | 0.003 | | | 0.88 | | | 0.29 |
| Boys | 3.00 (2.00, 5.00) | 19 | | −0.33 (−1.00, 0.00) | 19 | | −0.08 (0.17) | 33 | |
| Girls | 6.00 (4.00, 9.00) | 55 | | −0.33 (−0.60, 0.00) | 55 | | 0.11 (0.12) | 68 | |
| **Ethnicity** | | | 0.22 | | | 0.67 | | | 0.09 |
| White | 5.00 (3.00, 9.00) | 63 | | −0.33 (−0.63, 0.00) | 63 | | 0.06 (0.98) | 88 | |
| Other ethnicities* | 4.00 (3.00, 7.00) | 17 | | −0.07 (−0.86, 0.00) | 17 | | −0.28 (1.05) | 20 | |
| **Area SES (IMD)** | | | 0.20[a] | | | 0.63[a] | | | 0.10[a] |
| 1-5 (most deprived) | 5.50 (4.00, 8.00) | 30 | | −0.35 (−0.86, 0.00) | 30 | | −0.24 (1.00) | 42 | |
| 6-10 (least deprived) | 4.50 (3.00, 6.00) | 34 | | −0.33 (−0.60, 0.00) | 34 | | 0.09 (0.92) | 47 | |
| **Household SES (FAS)** | | | 0.71[a] | | | 0.87[a] | | | 0.02[a] |
| Low (0–7) | 5.50 (2.50, 9.00) | 24 | | −0.24 (−0.74, 0.00) | 24 | | −0.18 (1.18) | 29 | |
| High (8–14) | 5.00 (3.50, 8.00) | 56 | | −0.33 (−0.63, 0.00) | 56 | | 0.06 (0.93) | 79 | |

[a]p-value calculated using continuous variable

*Other ethnicities: Black African, Bangladeshi, Pakistani, Chinese, Afghan, British Indian, British Punjabi, Arabic, British Asian, Turkish, Mixed

Purchasing Healthfulness Scores closer to 1 indicate healthier purchasing

who were more affluent according to both area level SES (IMD) (p=0.10) and household SES (Family Affluence Score) (p=0.02).

**Aim 1: To assess the associations between food purchasing behaviours and diet quality in adolescents**

Table 4 presents the results of linear regression analyses, testing the association between food purchasing and diet quality. Unadjusted models showed no notable relationship between number of weekly food purchases and diet quality. There was also no notable relationship when models were adjusted for age, gender, ethnicity, and household SES. When the healthfulness of these purchases was considered, a positive association with diet quality was observed. Higher Weekly

**Table 4. Linear regression results for association between food purchasing (exposure) and diet quality (SDs) (outcome).**

| Exposure variable | Unadjusted β (95% CI) | p-value | Adjusted β (95% CI)* | p-value |
|---|---|---|---|---|
| Number of weekly purchases | 0.00 (−0.04, 0.04) | 0.98 | 0.01 (−0.04, 0.04) | 0.78 |
| Weekly Purchasing Healthfulness Score (healthfulness units) | 0.48 (0.01, 0.96) | 0.05 | 0.36 (−0.15, 0.87) | 0.16 |

*Models adjusted for age, gender, ethnicity, and household SES.

Purchasing Healthfulness Scores closer to 1 indicate healthier purchasing

Purchasing Healthfulness Scores were associated with higher Diet Quality Scores (β 0.48 SDs/healthfulness unit, (95%CI 0.01, 0.96) p = 0.05). These results were attenuated after adjustment (β 0.36 SDs/healthfulness unit, (95%CI −0.15, 0.87) p = 0.16).

**Aim 2: To assess whether SES moderates the relationship between food purchasing behaviours and diet quality**

Adding the interaction term to the regression models showed that the association between Weekly Purchasing Healthfulness Score and diet quality was stronger among participants with low household SES compared to those with high household SES (p for interaction = 0.01) (Fig 2). Stratified analyses, adjusted for age, gender and household SES, showed that less healthy weekly purchasing was associated with poorer diet quality among adolescents from households of lower SES (β 1.04 SDs/healthfulness unit, (95%CI 0.26, 1.81) p = 0.01). Among adolescents with higher household SES, no clear association was observed (β 0.27 SDs/healthfulness unit, (95%CI −0.25, 0.81) p = 0.31).

## Discussion

### Summary of key findings

This study provides novel insight into the independent food purchasing behaviours of adolescents and their relationship with overall dietary quality. The majority of food purchases made by adolescents participating in this study, when they were away from home and school, were not aligned with national healthy eating guidelines and had the potential to negatively impact their health. Adolescents in this study reported that 39% of their food and drink purchases were snacks rather than a main meal and these snack purchases were less healthful products according to the UK Eatwell Guide. Adolescents who made less healthful purchases tended to have poorer quality diets; this was particularly true for adolescents experiencing socio-economic disadvantage.

### Interpretation and comparison with previous research

Previous research has shown that adolescents source the majority of their food from home and school [6,8]. This current study highlights the important role of independent food purchases from food outlets, outside of home and school environments, on adolescents' dietary quality. This study also provides insights into the types of foods adolescents purchase for themselves and offers a more nuanced understanding than previous studies which have reported adolescent purchasing of only specific food items (e.g., sugar-sweetened beverages) or unspecific categories such as junk food or snack food [30,31]. While the majority (59%) of the purchases made by participants in this study did not adhere to the UK Eatwell Guide, adolescents did purchase, albeit in smaller quantities, some healthier food and drink items. Further research is needed to understand what motivates these healthier choices and how these factors might vary by SES.

Disparities in dietary behaviours between socioeconomic groups are well documented in the literature. Previous research has shown that adolescents from disadvantaged backgrounds tend to have poorer quality diets, with higher intakes of energy-dense, nutrient-poor foods and lower intakes of fruits and vegetables [32–34]. Additionally, previous

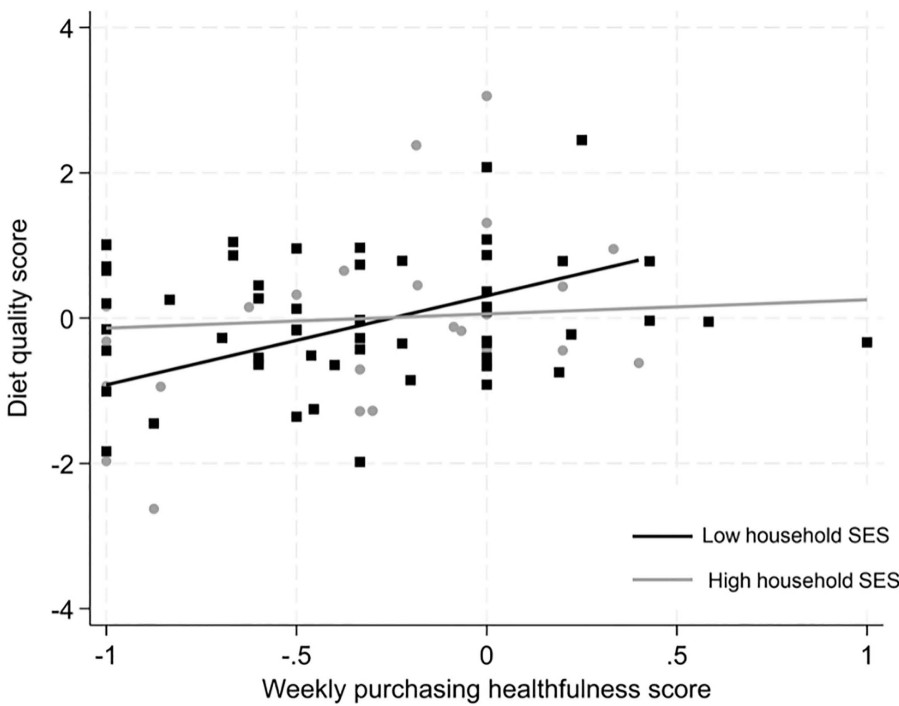

**Fig 2. Associations between diet quality scores and weekly purchasing healthfulness scores by household socioeconomic status (SES), based on linear regression analysis adjusted for age, gender, ethnicity, and household SES.**

research illustrates that food purchases made by adults experiencing socio-economic disadvantage are less healthy than those made by more affluent adults [35,36]. The findings from this study differ from those of previous studies, showing no variation in the number or healthfulness of purchases between adolescents from high and low SES households. Adolescents who purchased more healthful products, however, were more likely to have better quality diets. There was no clear relationship between the number of purchases made by adolescents and their diet quality. Together, these findings suggest that diet quality in adolescents is influenced more by the healthfulness of their purchases than by how often they shop.

Despite this research showing that on average all adolescents make more less healthy than healthy food purchases, this practice was shown to be most strongly associated with the dietary quality of adolescents with lower SES. One potential explanation for this finding is that independent food purchases make up a larger proportion of the overall diets of adolescents from more deprived backgrounds. Since these purchases tend not to conform to healthy eating guidelines, they are likely to reduce the overall quality of these young people's diets. Another driver of this difference may be that adolescents from less affluent backgrounds are not exposed to the same health-promoting foods at home as those living in more affluent households. Previous research conducted among Scottish families with teenagers found that even though health is valued when making food choices, families from lower SES were driven to make choices that ensured all family members were fed in a quick and acceptable way [37]. Backett-Milburn et al (2006) described how competing priorities within families with lower-incomes means nutrition and health fall below other factors such as affording everyday essentials, safety and worries about children engaging in other risky behaviours such as drugs, alcohol and smoking [38]. Additionally, a report published in 2023 further highlighted how the cost-of-living crisis has made healthy eating seem more unattainable for many low-income household due to rising food and energy costs [39]. This research has also illustrated that families experiencing low-income are driven to select less healthful foods because of poor housing and food preparation facilities, having less autonomy over their working practices and to fulfil a range of social, emotional

and cultural needs; financial constraints mean they are less able to fulfil these needs in other ways [39,40]. In contrast, results from a qualitative study have shown families experiencing higher SES have greater opportunity to offer experiences incorporating healthy foods and were more likely to discuss the nutritional significance of food choices in the home [37]. These families described limiting their intake of sugary and fatty foods while simultaneously encouraging younger family members to try a variety of different 'spicy' or 'exotic' foods to widen their taste preferences when they became adults [37]. Taken together, this body of evidence suggests that adolescents from lower SES do not have the same opportunities to consume a healthy or varied diet in the home as those from higher SES. Therefore, the purchases adolescents experiencing low SES make for themselves, when out of the home, may be particularly important targets for improving their quality of their diets.

## Research and policy implications

This study suggests that finding ways to promote and encourage more healthful independent food choices among adolescents, particularly those from disadvantaged backgrounds, is likely to be an important strategy in improving their overall dietary quality and, in turn, reducing dietary inequalities. Given the challenges faced by those from low-income households in relation to affording and acquiring heathy foods, the responsibility to support adolescents to eat more healthfully should not fall solely on parents but should be the responsibility of wider society. Future public health strategies and food policies will be most effective if they aim to support adolescents to acquire healthy food from multiple different sources, including neighbourhood and school environments.

Changing adolescents access to [41,42] and the environments inside the food outlets adolescents use most often could also help support more healthful purchases. The UK government have introduced restrictions on placing foods high in fat, sugar and salt in prominent areas of food stores. In addition, restrictions on the promotion of these items have been proposed [43]. Such food policies offer the opportunity to support more healthful purchases among all young people as long as they are implemented effectively across all types of retail outlets, neighbourhoods and regions [44].

Snack purchases made up the largest proportion of purchases by adolescents in the current study and tended to be less healthful. This finding is consistent with research from the Netherlands which showed adolescents mainly purchased snack food items from supermarkets close to school and spent, on average €2.30 (~USD 2.45) [45]. Qualitative research shows that adolescents favour quick, single-serve, low-cost snack options [46]. Future research and food policy development would benefit from targeting the availability of healthier snack options that are affordable and appealing to adolescents. These healthier snack options should be available in the food outlets adolescents regularly visit and offer a possibility for new product development.

## Strengths and limitations

This study is novel in its assessment of adolescents' independent food purchasing behaviours in the community setting. The EMA techniques used to collect purchasing data are a strength of the study because they allowed participants to record data as close as possible to the behaviour of interest, helping to reduce the risk of recall bias. In this study, it was not possible to determine the completeness of the purchasing data. Boys and younger adolescents were more likely to report no food purchases. Of those who did record purchases, girls and older adolescents tended to record higher quantities of food purchases during the study period. These patterns suggest that the absence of recorded purchases may reflect genuine lack of independent food purchasing rather than missing data. However, it is also possible that some participants who reported no purchases did not fully adhere to the study protocol. While the purchasing data provided a more detailed description of purchases made by adolescents than many published studies, data about food brands, portion sizes and nutritional information were not collected. It is therefore possible that purchases could have been miscategorised in terms of healthfulness. Where photos were available (23% of purchases), the research team used these to cross-check that participants had accurately categorised their food purchases. This cross-check was not possible when photos were not provided.

The study's small sample size reduces statistical power and the use of a convenience sample limits generalisability of the findings. The direction of the study findings, however, show promise and justify replication in a larger and more diverse population.

## Conclusion

This study demonstrated that adolescents predominately purchase food and drink items that are not aligned with healthy eating guidelines. Such purchasing decisions are more important for overall dietary quality among adolescents experiencing disadvantage. Findings from this study suggest that introducing food policies that support adolescents to make healthy food purchases from multiple different food environments (namely community and school) could act to reduce inequalities and improve health and well-being among the next generation of adults. Such policies may include zoning policies to limit the number of unhealthy food outlets in deprived areas, addressing placement within shops of unhealthy food promotions, widening access to free school meals and developing healthy, affordable and socially desirable snack options.

## Supporting information

**S1 Table. Coding of food purchases against the UK Eatwell Guide.**
(DOCX)

**S2 Table. Sensitivity analysis when purchases coded as '0' are treated as missing.** Linear regression results for association between food purchasing (exposure) and diet quality (SDs) (outcome).
(DOCX)

## Acknowledgments

The authors would like to thank the teachers and youth leaders who helped with the recruitment process for this study as well as all the young people who participated. The authors would also like to thank Patsy Coakley for her assistance in preparing the data for this study.

## Author contributions

**Conceptualization:** Sarah Shaw, Mary Barker, Christina Vogel.

**Data curation:** Sarah Shaw.

**Formal analysis:** Sarah Shaw, Sarah Crozier, Dianna Smith, Christina Vogel.

**Funding acquisition:** Sarah Shaw, Mary Barker, Christina Vogel.

**Investigation:** Sarah Shaw.

**Methodology:** Sarah Shaw, Christina Vogel.

**Project administration:** Sarah Shaw.

**Supervision:** Sarah Crozier, Cyrus Cooper, Dianna Smith, Mary Barker, Christina Vogel.

**Writing – original draft:** Sarah Shaw.

**Writing – review & editing:** Sarah Shaw, Sarah Crozier, Cyrus Cooper, Dianna Smith, Mary Barker, Christina Vogel.

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
