## [Decision Letter · Decision Letter 0]

8 Dec 2024

PONE-D-24-45297What role do adolescents’ independent food purchasing choices play in their dietary quality?PLOS ONE

Dear Dr. Shaw,

Thank you for submitting your manuscript to PLOS ONE. After careful consideration, we feel that it has merit but does not fully meet PLOS ONE’s publication criteria as it currently stands. Therefore, we invite you to submit a revised version of the manuscript that addresses the points raised during the review process.

We look forward to receiving your revised manuscript.

Kind regards,

Sandra Boatemaa Kushitor, Ph.D.

Academic Editor

PLOS ONE

Journal Requirements:

3. Thank you for stating the following financial disclosure: [This research and the authors of this paper are supported by the following funding sources: National Institute for Health Research Southampton Biomedical Research Centre; UK National Institute for Health Research Programme Grants for Applied Research (RP-PG-0216-20004); and UK Medical Research Council (MC_UU_12011/4). The views expressed in this publication are those of the author(s) and not necessarily those of the National Health Service, the National Institute for Health Research, and the UK Department of Health and Social Care.]. Please state what role the funders took in the study. If the funders had no role, please state: "The funders had no role in study design, data collection and analysis, decision to publish, or preparation of the manuscript." If this statement is not correct you must amend it as needed. Please include this amended Role of Funder statement in your cover letter; we will change the online submission form on your behalf.

4. Thank you for stating the following in your Competing Interests section: [SS, SC, CC, DS, MB have no conflicts of interests to declare. CV has a non-financial research collaboration with a UK supermarket chain. The study described in this manuscript is not related to this relationship.]. Please complete your Competing Interests on the online submission form to state any Competing Interests. If you have no competing interests, please state "The authors have declared that no competing interests exist.", as detailed online in our guide for authors at http://journals.plos.org/plosone/s/submit-now This information should be included in your cover letter; we will change the online submission form on your behalf.

5. In the online submission form, you indicated that [Due to ethical restrictions imposed in the interest of protecting participant confidentiality, the data underlying this study are available upon request. Researchers wishing to use the data can make apply to the research team by emailing mrcleu@mrc.soton.ac.uk. Subject to approval that the intended purpose is compatible with the study’s ethical approval and formal agreements regarding confidentiality and secure data storage being signed, the data would then be provided.].

6. Please amend your list of authors on the manuscript to ensure that each author is linked to an affiliation. Authors’ affiliations should reflect the institution where the work was done (if authors moved subsequently, you can also list the new affiliation stating “current affiliation:….” as necessary).

7. We note that you have referenced (unpublished data) on page 9, which has currently not yet been accepted for publication. Please remove this from your References and amend this to state in the body of your manuscript: (ie “Bewick et al. [Unpublished]”) as detailed online in our guide for authors

Additional Editor Comments:

Methodology

The authors should indicate which of the variables were normally distributed and which ones were not. It will also be helpful in the variables section for the authors to indicate how the variables were categorized. For example, readers are lost as to how the diet quality finally used in the analysis.

In table 5, it will be good to include the standard errors because the confidence intervals are quite wide. With a sample of size of 80 and some missing participants, the authors should include a section about their regression diagnostics.

Reviewers' comments:

Reviewer's Responses to Questions

**Comments to the Author**

1. Is the manuscript technically sound, and do the data support the conclusions?

Reviewer #1: Partly

2. Has the statistical analysis been performed appropriately and rigorously? 

Reviewer #1: No

3. Have the authors made all data underlying the findings in their manuscript fully available?

Reviewer #1: No

4. Is the manuscript presented in an intelligible fashion and written in standard English?

Reviewer #1: Yes

5. Review Comments to the Author

Reviewer #1: This study reports the results of study using ecological momentary assessment to measure indoendent food purchases among adolescents and assesses the association with dietary quality via FFQ. A strength is the use of EMA methods, however the convenience sample is very small and I have doubts about the suitability of the sample for many of the analyses presented and some of the analytic approach.

The introduction and discussion are mostly well written and well referenced. Perhaps the discussion is a little on the long side and could be trimmed down using more concise language.

Methods -

Line 48 - how were ppts recruited from these locations? Posters, email lists, announcements etc?

Line 64 "where possible, member of research team..." may be useful to report as a %of total sample how many times research team were present

Photographs were submitted for 23% of food entries, and photos / receipts were used to assess accuracy. What were the results of this checking process (ie how many of those photographs matched the ppts description of the food item?) Is there any other evidence to support that the participants were able to accurately and reliably categorise food items purchased?

Given the feasibility / acceptability results have not been published elsewhere, could the authors briefly summarise here? (Even if there are plans to publish this in future, a very high level summary would be useful)

The healthiness score tallied scores of -1 and 1 for non-adhering and adhering foods respectively. And then foods that could not be categorised were coded as 0 and included in this overall score. This approach essentially treats items of unknown "healthiness" as being halfway between the known values of -1 and 1. However, the more accurate treatment of these foods would be to code them as missing and not include them in the overall score. For example, a combination food "sandwich" might in reality fall in the "adhering" category. But defaulting to categorise it as "0" will pull the overall score downwards artificially. The opposite for a sandwich that would be accurately coded as "not adhering". I suggest the authors conduct the analyses coding these items as "missing" and present this as a sensitivity analysis, at the very least, otherwise replace analyses with the score recalculated treating uncategorised food as missing.

Line 151 - I'm not familiar with the term "test for trend" or how it retains statistical power (which test is this one replacing due to small numbers?)

Results-

What was the justification for the sample size recruited? Was an a priori power calculation conducted? Rationale is needed. In particular, I'm doubtful that the N provides sufficient power for the interaction test (maybe on the low side to address aim 1, even). Further, statistical testing of many of the subgroup differences presented in table 4 is not appropriate given the very small sample size. Further, in text, these results are described and statements about purchasing being "higher" in one group compared to another are based on comparison of the raw numbers. Despite p values being reported (which I don't think is appropriate for many of the presented analyses), they don't appear to have been used as a criterion for determining where differences lie. I'd suggest being very clear in the text that these comparisons are not supported by statistical tests due to small numbers within subgroups.

Given the study used a convenience sample, what was the rationale behind presenting adjusted analysis (including demographics that were unequally distributed across levels of the key outcome and exposure variables) as sensitivity analyses? It seems that failing to control for these variables is not optimal treatment of the data. I would encourage authors to present adjusted analyses as the primary analyses, and then report the unadjusted analyses as sensitivity analyses.

In tables and figures presenting healthfulness scores, it would be useful to specify as a reminder that scores closer to 1 indicate a healthier score, and closer to -1 indicate less healthy. This will assist readers viewing these elements out of context.

Line 228 "this relationship remained when.." this is a bit confusing given there was no significant relationship in the first place, suggest amending text to "there was also no significant relationship between x and y when..."

The conventional criteria for statistical significance is p less than .05, not equal to .05. Please amend the text in the results at line 232, Abstract, and Discussion accordingly.

The information presented in Figure 2 would be easily conveyed in text, so I would suggest deleting this figure for brevity.

I wasn't sure whether the results reported in text and in figure 3 were based on regression analyses adjusting for all other covariates or not. Please ensure this is clear- either way, from the methods it sounds as though both adjusted and unadjusted were run, but at present only one set of analyses are reported in the results.

Discussion

Line 372 "because of uncertainty regarding who made no recordings" - I'm struggling to grasp what this sentence means - I note that the following sentence provides further detail, but I'd suggest replacing them both with the following wording or similar: "...because it could not be determined whether participants who did not record any purchases did so because they did not make any independent food purchases or because they did not adhere to the study protocol"?

6. PLOS authors have the option to publish the peer review history of their article (what does this mean?). If published, this will include your full peer review and any attached files.

Reviewer #1: No

---

## [Author Response · Author response to Decision Letter 1]

2 Jun 2025

Please find a 'Response to Reviewer' letter attached

---

## [Editor Report · Decision Letter 1]

27 Jun 2025

PONE-D-24-45297R1What role do adolescents’ independent food purchasing choices play in their dietary quality?PLOS ONE

Dear Dr. Shaw,

Thank you for submitting your manuscript to PLOS ONE. After careful consideration, we feel that it has merit but does not fully meet PLOS ONE’s publication criteria as it currently stands. Therefore, we invite you to submit a revised version of the manuscript that addresses the points raised during the review process.

 Please address the comments I have raised in the attached file. 

We look forward to receiving your revised manuscript.

Kind regards,

Sandra Boatemaa Kushitor, Ph.D.

Academic Editor

PLOS ONE

Journal Requirements:

Additional Editor Comments:

Dear Authors,

Congratulations for revising the manuscript. The current format of the manuscript is well done, however the discussions and implications are not supported by the findings. I have made suggestions for you in the attached file for you to consider.

---

## [Author Response · Author response to Decision Letter 2]

17 Jul 2025

We thank the editorial team and reviewer for their helpful review of our manuscript and for providing us with valuable feedback.

We have address all the comments and attached a manuscript with tracked changes as well as a clean version. Our response to review letter outlines all the changes in detail.

---

## [Editor Report · Decision Letter 2]

24 Jul 2025

The association between adolescents’ independent food purchasing and dietary quality differs by socioeconomic status: findings from a pilot study

PONE-D-24-45297R2

Dear Dr. Shaw,

We’re pleased to inform you that your manuscript has been judged scientifically suitable for publication and will be formally accepted for publication once it meets all outstanding technical requirements.

Kind regards,

Sandra Boatemaa Kushitor, Ph.D.

Academic Editor

PLOS ONE
---

## [Editor Report · Acceptance letter]

PONE-D-24-45297R2

PLOS ONE

Dear Dr. Shaw,

I'm pleased to inform you that your manuscript has been deemed suitable for publication in PLOS ONE. Congratulations! Your manuscript is now being handed over to our production team.

Kind regards,

on behalf of

Dr. Sandra Boatemaa Kushitor

Academic Editor

PLOS ONE